# Projected Climate Could Increase Water Yield and Cotton Yield but Decrease Winter Wheat and Sorghum Yield in an Agricultural Watershed in Oklahoma

Solmaz Rasoulzadeh Gharibdousti [1,*], Gehendra Kharel [2], Ronald B. Miller [3], Evan Linde [4] and Art Stoecker [5]

1   Division of Agricultural Sciences and Natural Resources, Oklahoma State University, Stillwater, OK 74078, USA

2   Department of Natural Resource Ecology and Management, Oklahoma State University, Stillwater, OK 74078, USA; gehendra.kharel@okstate.edu

3   Department of Biosystems and Agricultural Engineering, Oklahoma State University, Stillwater, OK 74078, USA; ron.miller@okstate.edu

4   High Performance Computing Center, Oklahoma State University, Stillwater, OK 74078, USA; elinde@okstate.edu

5   Department of Agricultural Economics, Oklahoma State University, Stillwater, OK 74078, USA; art.stoecker@okstate.edu

*   Correspondence: rasoulz@okstate.edu; Tel.: +1-330-906-4988

**Abstract:** Climate change impacts on agricultural watersheds are highly variable and uncertain across regions. This study estimated the potential impacts of the projected precipitation and temperature based on the downscaled Coupled Model Intercomparison Project 5 (CMIP-5) on hydrology and crop yield of a rural watershed in Oklahoma, USA. The Soil and Water Assessment Tool was used to model the watershed with 43 sub-basins and 15,217 combinations of land use, land cover, soil, and slope. The model was driven by the observed climate in the watershed and was first calibrated and validated against the monthly observed streamflow. Three statistical matrices, coefficient of determination ($R^2$), Nash-Sutcliffe efficiency (NSE), and percentage bias (PB), were used to gauge the model performance with satisfactory values of $R^2 = 0.64$, NS = 0.61, and PB = +5% in the calibration period, and $R^2 = 0.79$, NSE = 0.62, and PB = −15% in the validation period for streamflow. The model parameterization for the yields of cotton (PB = −4.5%), grain sorghum (PB = −27.3%), and winter wheat (PB = −6.0%) resulted in an acceptable model performance. The CMIP-5 ensemble of three General Circulation Models under three Representative Concentration Pathways for the 2016–2040 period indicated an increase in both precipitation (+1.5%) and temperature (+1.8 °C) in the study area. This changed climate resulted in decreased evapotranspiration (−3.7%), increased water yield (23.9%), decreased wheat yield (−5.2%), decreased grain sorghum yield (−9.9%), and increased cotton yield (+54.2%) compared to the historical climate. The projected increase in water yield might provide opportunities for groundwater recharge and additional water to meet future water demand in the region. The projected decrease in winter wheat yield—the major crop in the state—due to climate change, may require attention for ways to mitigate these effects.

**Keywords:** SWAT; global climate models; Southern Great Plains; climate change; crop yield; surface runoff

## 1. Introduction

Impacts of climate change on agricultural production and water resources have been reported globally [1–5]. Ray et al. [6] found that more than 60% of the variability in crop yield in top global production regions is associated with climate, with both positive and negative responses noted depending upon geographic locations and irrigation applications [7–9]. For example, future climate change could increase corn and wheat yields in high latitudes and reduce them in middle to low latitudes [9,10]. Kang et al. [8] found that yields of wheat, rice, and maize were more sensitive to precipitation than temperature and generally increased with increased precipitation. On the other hand, Kang et al. [8] and others indicated increased crop production with a modest rise in average temperature of 1–3 °C, but decreasing yields above this range. From the hydrological modeling perspective, the Soil and Water Assessment Tool (SWAT) [11] has been used to assess quality and quantity issues [12,13] to identify critical source areas [14] and impacts on crop-yield [15,16] due to changes in climate and land uses in order to suggest improved management practices [17].

The Southern Great Plains in the U.S. is a water-limited and sometimes highly irrigated agricultural and oil-producing region which has experienced recurring droughts, which in turn have caused surface water losses and variability in crop yield [18]. Climate models project increased variability in precipitation and temperature for this region and thus suggest significant future variability in crop yield [19–21], as well as policy challenges related to the management of water required for food and energy-related economic interests [18,22].

Oklahoma is located in the Southern Great Plains, and similar to many states in that region, agriculture plays a key role in the state's economy. Therefore, understanding the effects of a changing climate on water resources and agricultural yields is crucial to developing sustainable and resilient mitigation measures. The water needs for crop irrigation, livestock, and aquaculture in Oklahoma amount to nearly 50% of the state's total water use and that percentage is projected to rise by 11–16% by 2060 [23,24]. It is projected that future precipitation and temperature will vary significantly in the state [25,26], but it is unknown if these changes will result in an increase or decrease in water and crop yields. For example, an increase in temperature in the region may increase crop water requirements leading to higher irrigation costs, stress on groundwater sources, and reduced profitability for farmers [27,28]. Alternately, increased precipitation or changes to seasonality may offset the impact of increased temperature, increasing crop yields. Understanding such changes is important in the water-stressed basins of central and western Oklahoma.

## 2. Materials and Methods

The Soil and Water Assessment Tool (SWAT) was used to develop a hydrological and agricultural production model of a portion of the Fort Cobb agricultural watershed in southwest Oklahoma. The model was calibrated and validated for monthly streamflow and annual yields of cotton, grain sorghum, and winter wheat. Future climate projections from three global climate models (GCMs) were then used as climate inputs for the validated model to estimate the climate-associated changes on watershed hydrology and crop yield. The sequence of model development and validation steps is illustrated in Figure 1, and described in detail in the following sections.

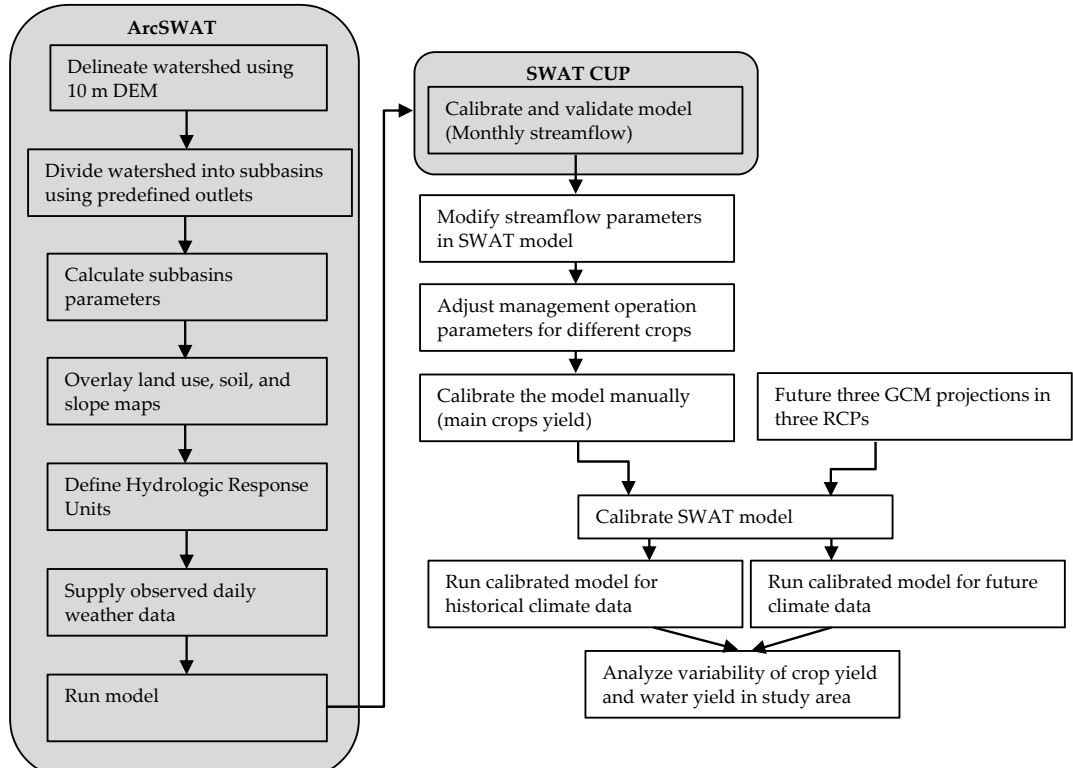

**Figure 1.** Conceptual model of integrated SWAT and climate model. DEM: Digital Elevation Model; GCM: General Circulation Model; and RCP: Representative Concentration Pathway.

### 2.1. Study Area

The study area consists of two sub-watersheds, Cobb Creek and Five Mile Creek, which are headwaters of the Fort Cobb Reservoir watershed (Figure 2). These two sub-watersheds were integrated into a single study area. Between 1982 and 2015, the study area received an annual average 2.2 mm/day precipitation, with a daily average temperature of 15.8 °C. The closest available streamflow gauge station maintained by the United States Geological Survey (USGS 07325800) receives runoff from these two sub-watersheds, and therefore provides an opportunity to calibrate and validate the hydrological model with observed streamflow data. These two sub-watersheds occupy an area of 342.6 km$^2$, 43% of the Fort Cobb Reservoir watershed, and the combined land use is approximately 50% cropland, 44% pastureland, and 6% other land cover types [29].

The major crops in the study area are winter wheat (34%), cotton (9%), and grain sorghum (1.5%); other crops grown in the study sub-watershed include alfalfa, canola, corn, and soybean (5.5%). Nearly half of the soils in the study region are predominantly silty, with lesser hydraulic conductivities. Field reconnaissance of the watershed revealed that a few of the older solid-set or side-roll irrigation systems are still used in the watershed, but that most irrigation systems have been upgraded to center-pivot systems from the Rush Springs aquifer [30]. Population in south-western Oklahoma, in which the study sub-watersheds are located, is sparse and decreasing. Agriculture focuses on commodity production (beef, wheat, and row crops) with high costs and low margins [31].

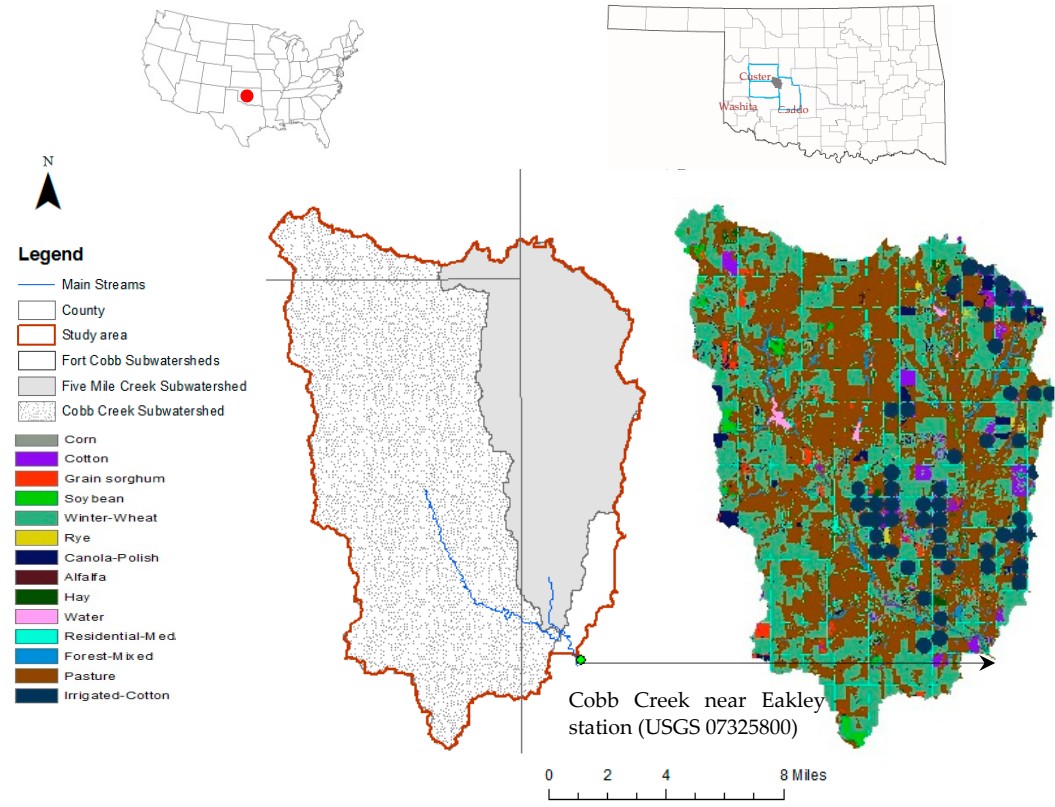

**Figure 2.** Five Mile Creek and Cobb Creek sub-watersheds within Forb Cobb Reservoir watershed located in three counties of Southwestern, OK, USA.

## 2.2. Hydrological Model

The SWAT model [11] was used to develop a hydrological model of the Fort Cobb study area, consisting of the Cobb Creek and Five Mile Creek sub-watersheds. SWAT is a hydrological modeling tool widely used to simulate the long-term effects of changes in climate, land use management, and agricultural practices [17,25,32,33]. The 10 m United States Geological Survey (USGS) Digital Elevation Model (DEM) was used to delineate the watershed boundary using the location of the USGS gauge station at Cobb Creek near Eakley (07325800) as the watershed outlet. The study area was then divided into 43 sub-basins with an average area of 8 km$^2$ (min. 0.2 km$^2$ and max. 28 km$^2$). Soil attributes including texture and moisture capacity were derived from the Soil Survey Geographic Database (SSURGO) [34]. The crop data layer for the year 2014 [29] was used to identify the locations of each crop and land cover type in the study watersheds. These soil and crop data and slope derived from the DEM were combined to produce 15,217 hydrologic response units (HRUs), which represent unique combinations of soil, land use, and slope. In SWAT, an HRU is the finest scale of measurement where routings of water, nutrients, and sediments are calculated and then aggregated to the sub-basin and the watershed level. A large number of HRUs were generated in our study at the expense of computational efficiency to understand the detailed effects of climate change on this agriculture-pasture intensive watershed.

The historical climate over the period 1982–2016 was characterized using data from two weather stations located in the study area and maintained by the Agricultural Research Service of the United States Department of Agriculture (https://datagateway.nrcs.usda.gov/). Daily averages for three weather variables, precipitation, minimum temperature, and maximum temperature, were collected and assigned to HRUs based on proximity to the weather station.

Data on crop management practices, including fertilizer application, management practices in the study area, and tillage type for the selected crops were obtained through consultation with local personnel from the Oklahoma State University Cooperative Extension Service and Conservation

District, and from the available literature [35,36]. Details about the management practices and how they were applied to the model are included in Appendix A (Table A1). Grazing is a significant factor in the watershed with an effect on hydrologic response, and therefore it is included in the model using the cattle stocking rate (0.5 head/ha), consumed biomass (3 kg/ha/day), trampled biomass (0.47 kg/ha/day), and deposited manure (1.5 kg/ha/day) as obtained from the USDA-NASS and Storm et al. [35,37].

Model Calibration and Validation

Streamflow observations recorded at the USGS gauge station 07325800 were obtained. The years 1991–2000 were used as the calibration period, while 2001–2010 was used as the validation period for the model (Figure 2). Seventeen parameters related to streamflow were manipulated using the automated SWAT Calibration and Uncertainty Procedures (SWAT-CUP) at monthly scale [38] (Table 1).

**Table 1.** Model parameters with the calibrated values for streamflow in the study area.

| Parameter | Description | Calibrated Value |
|---|---|---|
| GWQMN | Threshold depth of water in the shallow aquifer required for return flow to occur (mm) | 0.6 |
| GW_REVAP | Groundwater "revap" coefficient (unit less) | 0.02 |
| REVAPMN | Threshold depth of water in the shallow aquifer for "revap" to occur (mm) | 1.4 |
| RCHRG_DP | Deep aquifer percolation fraction (unit less) | 0.47 |
| GW_DELAY | Groundwater delay (days) | 376 |
| CN2 | SCS Curve number adjustment for soil moisture condition II (unit less) | −12.7 % of default values |
| ALPHA_BF | Baseflow Alpha Factor (days) | 0.95 |
| ESCO | Soil evaporation compensation factor (unit less) | 0.83 |
| EPCO | Plant uptake compensation factor (unit less) | 0.3 |
| CH_K1 | Effective hydraulic conductivity in tributary channel alluvium (mm/h) | 0.093 |
| SURLAG | Surface runoff lag coefficient (unit less) | 3.1 |
| EVRCH | reach evaporation adjustment factor (unit less) | 0.34 |
| TRNSRCH | Fraction of transmission losses partitioned to deep aquifer (unit less) | 0.095 |
| ALPHA_BNK | base flow alpha factor for bank (days) | 0.84 |
| SOL_AWC | Available water capacity of soil layer (mm $H_2O$/mm soil) | 0.036 |
| CH_N2 | Manning's n value for the main channel (unit less) | 0.18 |
| CH_K2 | Main channel conductivity (mm/h) | 1.98 |

The model performance for streamflow was evaluated using three statistical measures: coefficient of determination ($R^2$), Nash-Sutcliffe efficiency (NSE), and percentage bias (PB). The values of $R^2$ (0.64), NSE (0.61), and PB (5%) (Figure 3) in the model calibration period were deemed to be satisfactory by metrics suggested by other SWAT-based studies [39,40]. The calibrated model was then validated by comparing the USGS observations with SWAT simulated streamflow for the ten-year time period 2001–2010. The model performance with the validation dataset ($R^2$ = 0.79; NSE = 0.62; PB = −15%) was reasonable, as shown in Figure 3.

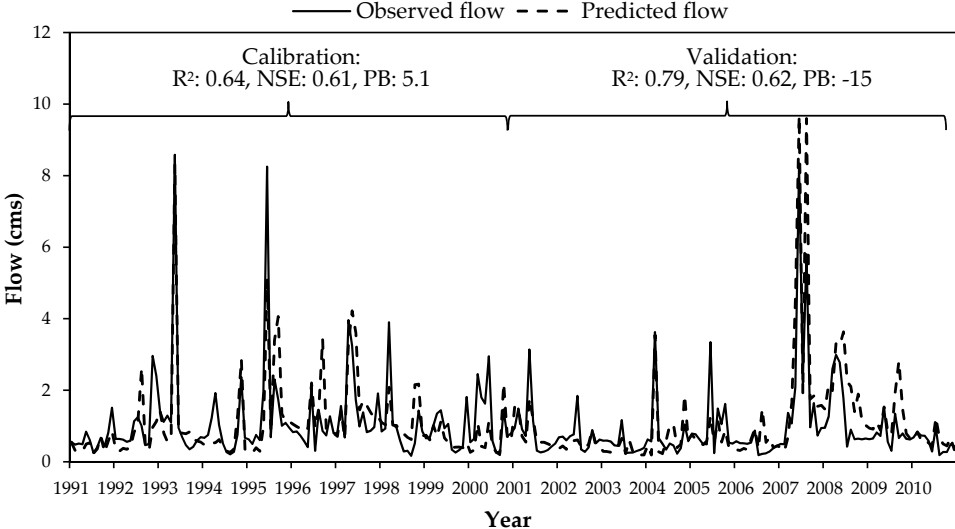

**Figure 3.** Average monthly observed and simulated streamflow for the calibration (1991–2000) and validation (2001–2010) periods.

Manual calibration of the model for yields of cotton, grain sorghum, and winter wheat followed successful hydrologic calibration using percent bias as a measure of performance, which is popularly used in crop-yield modeling studies [41,42]. Yield data were based on the Oklahoma State University experimented variety trials data for the years 2005 to 2010 (http://croptrials.okstate.edu/) and county level crop yield data for the years 1986 to 2005 (Oklahoma Agricultural Statistics database, http://digitalprairie.ok.gov/cdm/ref/collection/stgovpub/id/11177). The variety trial crop yield data came from experimental sites in seven southwestern Oklahoma counties (Blaine, Caddo, Canadian, Comanche, Custer, Grady, and Tillman), including those where the study watershed is located (Figure 1). Ten crop model parameters were selected (Table 2) and their associated value ranges were set based on recommendations made by Nair et al. [43]. The values were then manually adjusted until the PB for the crop models reached satisfactory values for cotton (−4.5%), grain sorghum (−27.3%), and winter wheat (−6.0%) from the year 1986 to 2010 (Figure 4).

**Table 2.** Cotton, grain sorghum, and winter wheat yield calibration parameters.

| Parameter | Unit | Parameter Definition | Calibrated Values | | |
|---|---|---|---|---|---|
| | | | Cotton | Grain Sorghum | Winter Wheat |
| BIO_E | kg/ha/MJ/m$^2$ | Radiation use efficiency or biomass energy ratio | 14 | 37 | 29 |
| USLE_C | no unit | Minimum value of USLE C factor for water erosion | 0.1 | 0.2 | 0.02 |
| HVSTI | kg/ha/kg/ha | Harvest index for optimal growing season | 0.3 | 0.3 | 0.3 |
| OV_N | no unit | Manning's "n" value for overland flow | 0.12 | 0.12 | 0.12 |
| BLAI | m$^2$/m$^2$ | Maximum potential leaf area index | 3 | 4.5 | 3 |
| FRGRW1 | fraction | Fraction of plant growing season to the first point on the optimal leaf area development curve | 0.14 | 0.15 | 0.03 |
| FRGRW2 | fraction | Fraction of plant growing season to the second point on the optimal leaf area development curve | 0.3 | 0.5 | 0.35 |
| LAIMX1 | fraction | Fraction maximum leaf area index to the first point on the optimal leaf area development curve | 0.005 | 0.05 | 0.03 |
| CNYLD | kg N/kg seed | Normal fraction of nitrogen in yield | 0.018 | 0.02 | 0.02 |
| CPYLD | kg P/kg seed | Normal fraction of Phosphorus in yield | 0.0027 | 0.0032 | 0.0018 |

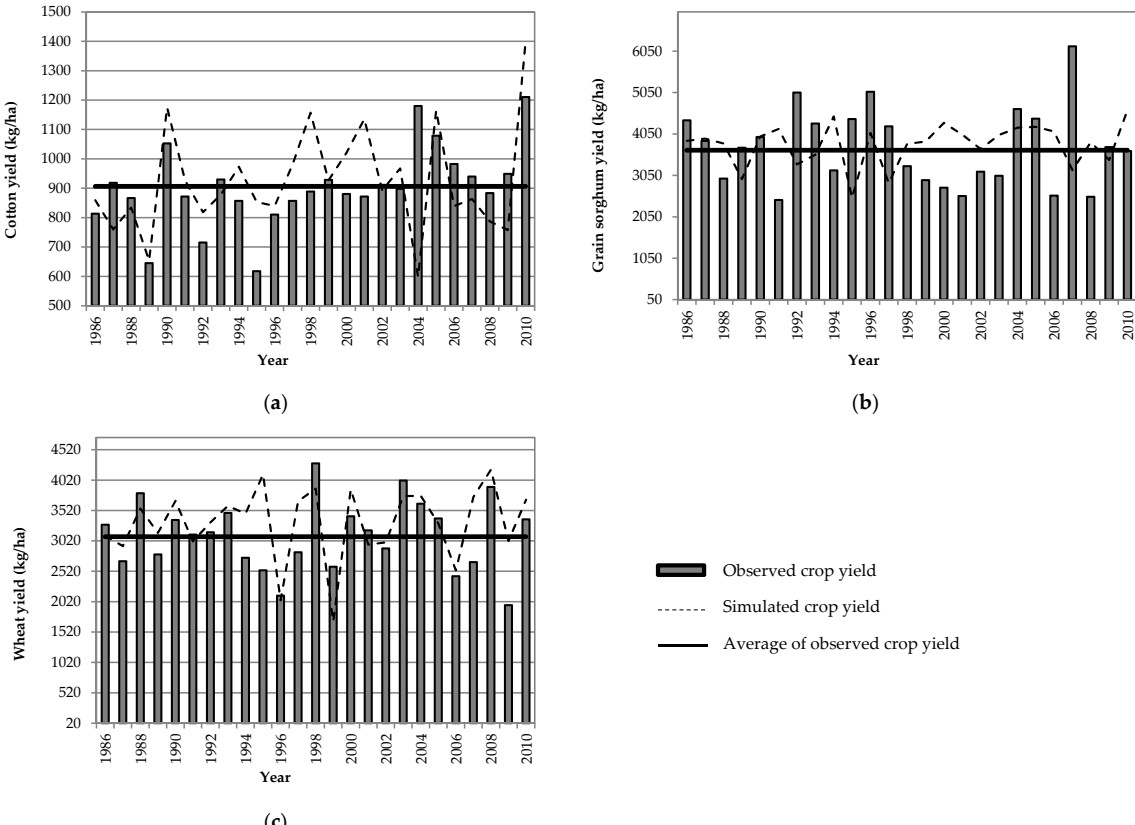

**Figure 4.** Observed and simulated average annual yields of (**a**) cotton, (**b**) grain sorghum, and (**c**) winter wheat in the study area.

### 2.3. Future Climate Data

Future climate projections for the study area were obtained from three GCMs (MPI-ESM-LR, CCSM4, and MIROC5) specifically downscaled for the Southern Great Plains where the study watersheds are located by the USGS–South Central Climate Science Center (SCCSC) (http://dx.doi.org/10.15763/DBS.SCCSC.RR). The SCCSC, in an effort to provide seamless climate projection data for the U.S. South Central region, downscaled climate data for the entire Red River Basin for three GCMs using three statistical downscaling methods with a spatial resolution of ~11 km [44]. Three GCMS are included in the Coupled Model Intercomparison Project-5 (CMIP-5) climate projections under the climate-forcing effects of three Representative Concentration Pathways: RCP 2.6, 4.5, and 8.5 (Table 3). The 3 GCMs and 3 RCPs created 9 climate scenarios for this study. Mendlik and Gobiet [45] suggested that an ensemble of multiple GCMs and RCPs be used for hydrological modeling to minimize the biases and uncertainties associated with GCM projections.

**Table 3.** The Coupled Model Intercomparison Project-5 global climate models used in the study.

| GCMs | Model Agency | Atmospheric Resolution (Lat × Lon) | Downscaled Resolution (Lat × Lon) | Downscaling Method |
|---|---|---|---|---|
| CCSM4 | National Center for Atmospheric Research, Boulder, CO, USA | 0.90 × 1.25 | | |
| MIROC5 | Atmosphere and Ocean Research Institute, University of Tokyo, Tokyo, Japan | 1.41 × 1.41 | 0.1 × 0.1 | Quantile mapping method-cumulative density function transform [44] |
| MPI-ESM-LR | Max Planck Institute for Meteorology, Hamburg, Germany | 1.80 × 1.80 | | |

Lat × Lon means: Latitude × Longitude.

## 3. Results

### 3.1. Future Climate

Nine combinations of the CMIP-5 climate projections (3 GCMs × 3RCPs) were generated for the future period 2016–2040. The nine CMIP5 climate scenarios indicated changes for the study area in average annual precipitation (−10.6% to +13.2%), and average annual temperature (+1.7 °C to +2.0 °C), compared to the 1986–2010 historical climate (Figure 5). The overall average of all nine future climate combinations indicated an increase in average annual precipitation of +1.5%. Examining the future trends of precipitation by RCP, on average the RCP2.6 indicated the greatest increase compared to the historical average (+2.9%), followed by RCP4.5 (+0.8%) and RCP8.5 (+0.6%). However, there were large differences in precipitation between the GCMs. For instance, RCP8.5 had the lowest average annual increase in precipitation but the highest variation among GCMs, with MPI-ESM-LR showing the highest increase (+13.1%) and CCSM4 showing the most reduced precipitation (−10.7%). Examining overall monthly averages, precipitation increased over the historical average markedly in February (+52.4%), April (+33.3%), and November (+23.8%) while it decreased in June (−27.5%), August (−11.3%) and December (−8.3%) (Figure 5a).

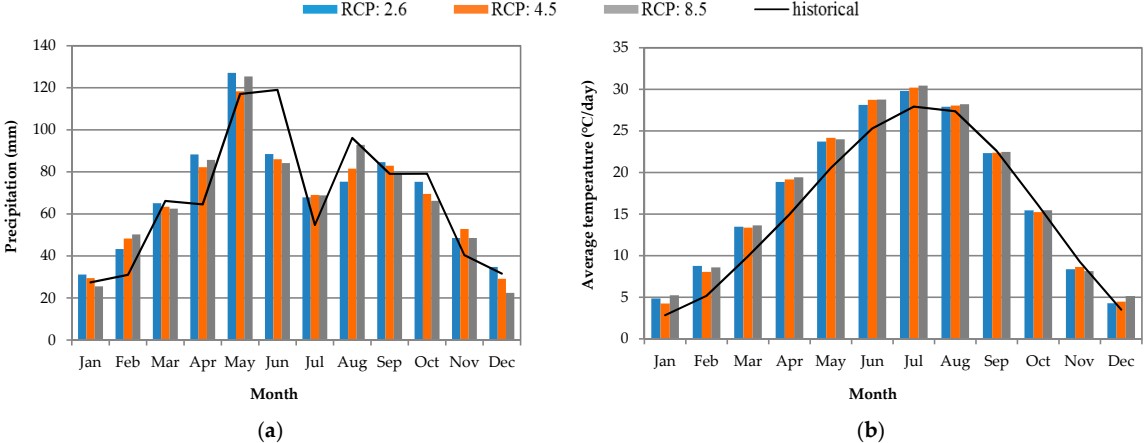

**Figure 5.** Average monthly precipitation (**a**) and average monthly temperature (**b**) under the historical (1986–2010) and future (2016–2040) climate. For future climate, values are presented as an average of all three GCMs for each climate scenario (RCP).

The average of all nine future climate combinations indicated an increase in average annual temperature of 1.8 °C. The RCP8.5 and RCP4.5 scenarios showed the highest average annual increase (+2.0 °C), followed by RCP2.6 (+1.7 °C). Within the RCP averages, the highest average annual temperature increase occurred in RCP 8.5 (CCSM4, +2.3 °C). Examining the overall monthly averages, the future climate indicated an increase in average monthly temperature during the first half of the year, with the highest increase in April (+4.2 °C) and the lowest increase in August (+0.7 °C), and average decreases in September (−0.2 °C), October (−0.7 °C), and November (−0.9 °C) (Figure 5b).

### 3.2. Water Balance

Model simulations showed that the projected changes in precipitation and temperature in the watershed would decrease potential evapotranspiration (PET) by 13.4% and actual evapotranspiration (ET) by 3.7%, leading to an overall increase in modeled water yield of 23.9% (Table 4). Water yield is the average volume of water reaching the watershed outlet over the total modeling period, and is reported as the unit length of water (volume/area). Examining the RCP averages, the greatest water yield increase occurred in RCP4.5 (148.4 mm/year), which also saw the lowest annual ET (693.3 mm/year) among the RCPs (Table 4). Overall, we found a markedly higher variation in water yield (−39.3% to

+92.8%) with MPI-ESM-LR and CCSM4 under RCP8.5, indicating the highest (+92.8%) and lowest (−39.3%) water yield respectively, compared to the historical period (Table 5).

**Table 4.** Average annual rainfall, potential evapotranspiration (PET), evapotranspiration (ET), surface runoff (SURQ), groundwater flow to stream (GWQ), and water yield (WYLD) under historical (1985–2010) and future climate (2016–2040). Water yield is the net amount of water entering the stream from each hydrologic response unit, groundwater flow and surface runoff.

| Climate Scenario | Rainfall | PET | ET | SURQ | GWQ | WYLD |
|---|---|---|---|---|---|---|
| | (mm) | (mm) | (mm) | (mm) | (mm) | (mm) |
| RCP2.6 Mean | 829.8 | 1604.6 | 709.5 | 32.4 | 48.7 | 142.6 |
| RCP4.5 Mean | 812.8 | 1676.5 | 693.3 | 31.7 | 51.9 | 148.4 |
| RCP8.5 Mean | 811.1 | 1706.6 | 704.4 | 30.3 | 45 | 133.4 |
| **Overall Mean** | **817.9** | **1662.6** | **702.4** | **31.5** | **48.6** | **141.4** |
| Modeled Historical Mean | 806.2 | 1920.1 | 729.2 | 38.3 | 30.7 | 114.2 |
| Percent change | 1.5% | −13.4% | −3.7% | −17.9% | 58.4% | 23.9% |

**Table 5.** Future water yield (WYLD) as modeled by future climate compared to the historical yield.

| RCP | GCM | WYLD (mm) | Change from Historical (%) |
|---|---|---|---|
| | MPI-ESM-LR | 193.5 | 69.5 |
| 2.6 | MIROC5 | 124.9 | 9.4 |
| | CCSM4 | 113.5 | −0.6 |
| | MPI-ESM-LR | 196.0 | 71.6 |
| 4.5 | MIROC5 | 136.0 | 19.1 |
| | CCSM4 | 118.6 | 3.9 |
| | MPI-ESM-LR | 220.2 | 92.8 |
| 8.5 | MIROC5 | 115.1 | 0.8 |
| | CCSM4 | 69.3 | −39.3 |
| Historical | | 114.2 | - |

Seasons are critically important for crop production, and therefore to help understand changes at a finer temporal scale, modeled values were calculated for Winter (December, January, February), Spring (March, April, May), Summer (June, July, August), and Autumn (September, October, November). On average, water yield increased relative to historical values in all four seasons with the highest increase in winter (+54.5%), followed by spring (+34.9%), fall (+24.6%), and summer (+14.8%) (Figure 6d).

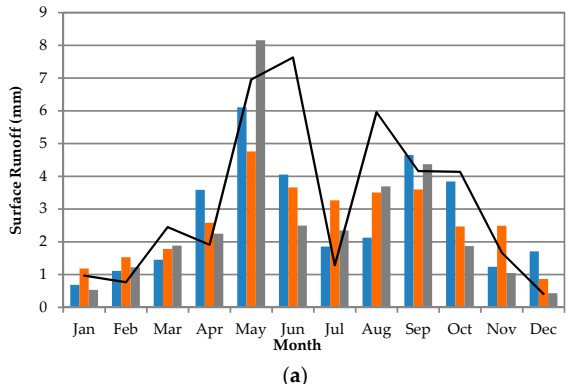

(a)

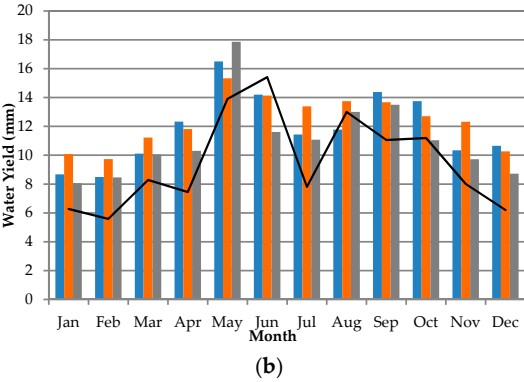

(b)

**Figure 6.** *Cont.*

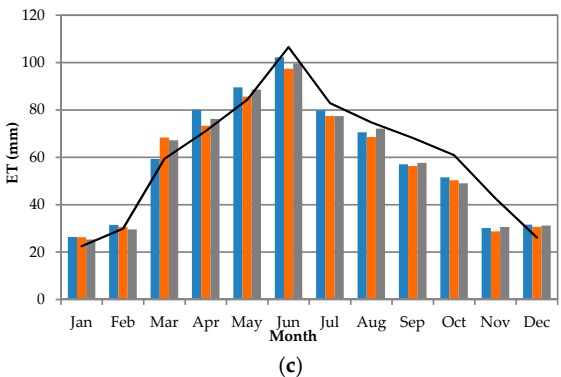
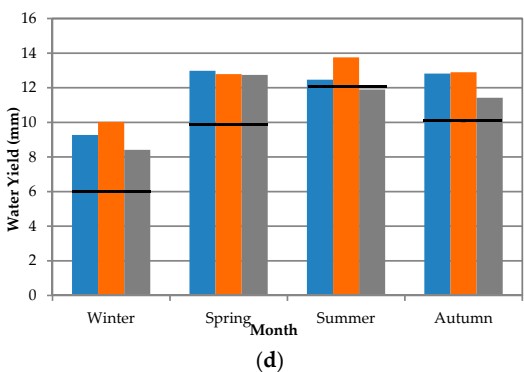

**Figure 6.** Model simulated (**a**) total monthly surface runoff, (**b**) total monthly water yield, (**c**) total monthly evapotranspiration, and (**d**) seasonal total water yield under historical (1986–2010) and future (2016–2040) climate. The values are presented as an average of all three global climate models (GCMs) for each representative concentration pathways (RCP).

### *3.3. Crop Yield*

The overall mean of climate scenarios indicated decreased yields for winter wheat (−225.07 kg/ha, −5.2%) and sorghum (−433.24 kg/ha, −9.9%), and increased yields for cotton (328.67 kg/ha, +54.2%) for the period 2016–2040 relative to the historical mean annual yields (Table 6). The modeled future winter wheat yield varied between −23.0% and +5.8%, with the greatest decline in MPI-ESM-LR/RCP 8.5 and the greatest increase in CCSM4/RCP 2.6 (Table 6). Of the nine future climate scenarios modeled, the winter wheat yield decreased in five. It was found that sorghum yields were consistently lower (−16.5% to −1.9%) and cotton yields higher (18.2% to 105.9%), compared to the historical yield in all nine climate scenarios.

**Table 6.** Percentage crop yield changes under the modeled climate change scenarios relative to the historical yield. The values represent the difference between the calculated average crop yield for the climate scenario and the observed average yield expressed as a percent of the observed yield.

| Crop | RCP 2.6 | | | | RCP 4.5 | | | | RCP 8.5 | | | | Overall |
|---|---|---|---|---|---|---|---|---|---|---|---|---|---|
| | MPI-ESM-LR | MIROC5 | CCSM4 | Mean | MPI-ESM-LR | MIROC5 | CCSM4 | Mean | MPI-ESM-LR | MIROC5 | CCSM4 | Mean | |
| Cotton | 73.1 | 18.2 | 33.9 | 41.7 | 103.4 | 28.8 | 32.8 | 55.0 | 105.9 | 27.3 | 64.7 | 66.0 | 54.2 |
| Sorghum | −1.9 | −8.3 | −8.8 | −6.3 | −8.1 | −13.0 | −16.5 | −12.5 | −2.4 | −16.3 | −13.4 | −10.7 | −9.9 |
| Winter wheat | −16.8 | 2.5 | 3.3 | −3.7 | −23.0 | −0.5 | 5.8 | −5.9 | −18.7 | 2.8 | −2.6 | −6.2 | −5.2 |

The majority of cotton grown in the study area is irrigated by center pivot, but a small percentage is dryland. This separation was included in the model through analysis of NASS aerial imagery (https://datagateway.nrcs.usda.gov/) and calculating the area of cotton plantings within and outside the center pivot circles, then only irrigating the actual center pivot area. Analysis of the climate scenario modeled cotton production shows differences between irrigated and dryland yield rates (Table 7). The overall irrigated cotton yields (1120.7 kg/ha) are significantly greater ($p = 0.006$) than the dryland yields (807.7 kg/ha).

**Table 7.** Mean annual irrigated and dryland cotton yields for future climate scenarios (2016–2040). The irrigated average annual yields are significantly greater than the dryland mean yields ($p = 0.006$).

| RCP | GCM | Cotton Yield (kg/ha) | | | |
|---|---|---|---|---|---|
| | | Irrigated | Mean | Dry Land | Mean |
| **2.6** | CCSM4 | 1015.7 | | 671.9 | |
| | MIROC5 | 804.1 | 1014.8 | 656.8 | 752.7 |
| | MPI-ESM-LR | 1224.6 | | 929.4 | |
| **4.5** | CCSM4 | 976.2 | | 688.1 | |
| | MIROC5 | 1018.2 | 1147.5 | 618.9 | 797.6 |
| | MPI-ESM-LR | 1448.1 | | 1085.7 | |
| **8.5** | CCSM4 | 1257.7 | | 820.9 | |
| | MIROC5 | 918.3 | 1199.9 | 671.5 | 873.5 |
| | MPI-ESM-LR | 1423.6 | | 1128.1 | |
| Overall Mean | | 1120.7 | | 807.9 | |

## 4. Discussion

The potential future changes in precipitation and temperature and how they interact to affect the water balance in agricultural watersheds is crucial. Overall, the nine climate scenarios based on three downscaled CMIP5 climate projections indicated increases in both precipitation and temperature for the study watershed over the 2016–2040 period modeled. The overall modeled increase in precipitation of nearly 1.5%, could mean increased water availability and opportunity for groundwater recharge (Table 4). Our estimates of increased precipitation in the study area are similar to the values estimated by Qiao et al. [26] who used a 39-member ensemble of the downscaled CMIP5 climate projections for the Arkansas Red River Basin, but differs from the estimates by Garbrecht et al. [18] of decreased precipitation in the region. However, the projected increase in average annual temperature of 1.9 °C in the watershed is similar to the increase Garbrecht et al. [18] estimated. The diverging precipitation estimates between the Garbrecht et al. [18] and our study could be due to the use of two different climate datasets; we used CMIP-5 projections while CMIP-3 was used by Garbrecht et al. [18]. Compared to CMIP-3, the CMIP-5 models include an improved physical representation and integration of the processes in the atmosphere, ocean, and land with higher resolution and a new representation of anthropogenic forcing of climate [46,47]. It was found that compared to CMIP-3 simulations, CMIP-5 ensembles have improved regional-scale temperature distributions with no systematic change for precipitation [46].

The GCMs in RCP 8.5 had the largest range in water yield, with values from +92.7% to −32.7% of the modeled historical yields. This extreme range in water yields (Table 5) appears to be related to rainfall, which had its highest value in MPI-ESM-LR and lowest in CCSM4 relative to the historical climate (Table 4). The overall increase in water yield occurred despite a significant reduction in average annual surface runoff (−17.9%), and thus is contributed entirely by increase in groundwater recharge (+58.4%) which has likely been influenced by a reduction in modeled actual ET (Table 4). The relatively low water yield increase in summer seen in Figure 6d could be due to a combination of higher temperature (+2.1 °C) and reduced precipitation (−4.6%) in the months of June and August compared to the historical climate (Figure 5a,b). Our finding of increased water yield in the watershed is similar to findings reported from other watersheds in the U.S. For example, in an agriculturally intensive watershed in the northern Great Plains, Neupane et al. [48] found climate change increased average water yield by 8–67%. Similar to modeled results from this study, Neupane et al. [48] reported that the increased water yield was due primarily to increases in groundwater contribution. Gautam et al. [15] reported a 29% increase in median water yield in a heavily agricultural experimental watershed of Missouri, USA under the CMIP-5 climate.

Crop modeling using the climate scenarios decreased winter wheat and grain sorghum yields and increased the yield of cotton in the watershed (Table 5). An increase in temperature of 2.9 °C

and relatively unchanged precipitation in the critical growing season could have led to lower soil moisture and suppressed winter wheat production. Our result of decreased winter wheat yield is consistent with the estimates of Rosenzweig et al. [9] and Delphine et al. [10], who found that wheat yield would decrease in low to mid-latitude areas of the globe due to climate change. Winter wheat is Oklahoma's most valuable crop and decreased wheat yield is of important concern because of its economic significance locally in the watershed and regionally. Nearly one third of the study watershed is traditionally planted with winter wheat, and Oklahoma is the fourth largest producer of winter wheat in the U.S.

Cotton production is sensitive to temperature, and according to Adhikari et al. [16] cotton yields in the Texas High Plains increased with temperature and sufficient water under future climate projections, including increased atmospheric $CO_2$. Their results indicated that the increased cotton yield could be partly attributed to increased temperature in the future, and that with additional atmospheric $CO_2$, cotton could potentially withstand the impacts of future climate variability if irrigation water remains available at current levels. In this modeling study, we allowed cotton irrigation at historical levels throughout the future simulation and thus, similar to Adhikari et al. [16], well-irrigated cotton was able to benefit from increased temperature. The modeled dryland cotton yields (807.9 kg/ha) were much smaller than the irrigated crop (Table 6) and showed no essential change from the modeled historical yield (808.3 kg/ha). Therefore, the potential climate benefits for future cotton production in the study area depend on sustainable management of water resources for irrigation.

This study represents an important first step towards understanding and adapting to the uncertainty that projected change in climate poses to an agricultural watershed. Projecting downscaled future climate scenarios onto the current mix of agricultural practices produces an understanding of future changes based on a familiar frame of reference, which is among the types of information needed by stakeholders such as agricultural advisors to alert local farmers of the need to adapt. Travis and Huisenga [49] found that the occurrence of extreme climate events increased the rate of adaptation to changing climate among farmers, which implies action after poor yields. Schattman et al. [50] noted that farmers perceive climate change risks in terms of known experience, and therefore are more likely to respond to adaptation planning information that incorporates typical activities.

Our study has important limitations that need to be addressed in future research. The first limitation is related to modeling of sediment and nutrient loadings which we excluded in this study but are important in the selection and implementation of best management practices in agricultural watersheds. Any changes in climate, hydrological response, and/or on-farm practices will likely change the yields of nutrients or sediments, and therefore should be studied. The next limitation is related to hydrological and climate model related uncertainty in the estimated water and crop yield; in this study we examined climate model related uncertainty by including an ensemble of three GCMs and three RCPs as suggested by Brown et al. [51], which yielded a general understanding of potential hydrologic and standard crop yield changes. The next step would be to utilize a sophisticated crop yield model, in the manner of Adhikari et al. [16] and Bao et al. [52], and at a regional scale with several sources and a mix of important model input and management scenarios as suggested by Daggupati et al. [12] and Akkari and Bryant [53], to better understand and prepare potential farm adaptation strategies at local and regional scale.

## 5. Conclusions

In this study, we investigated climate change impacts on surface runoff, water yield, and crop yield in an agricultural watershed of Oklahoma using a SWAT based hydrological model. We found that the study area saw increases in future average annual precipitation (1.5%) and temperature (1.9 °C) compared to the 1986–2010 climate. There was higher variability in precipitation between the GCMs with some indicating decreased precipitation, while others projected increased precipitation. These changes in precipitation and temperature led to decreased potential evapotranspiration (−13.4%) and evapotranspiration (−3.7%) resulting in an overall increase in water yield (+23.9%) in the study area.

The projected increase in water yield might provide opportunities for groundwater recharge and additional water to meet projected water demand in the region. With the future climate projections, the models simulated reduced yields for grain sorghum and winter wheat while the cotton yield increased significantly. The projected decrease in yield of winter wheat—the major crop in the watershed and in the state—due to climate change may require additional research on ways to mitigate these effects.

**Author Contributions:** Conceptualization, S.R.G. and A.S and G.K.; methodology, S.R.G. and G.K.; software, S.R.G.; validation, S.R.G. and G.K.; formal analysis, S.R.G., G.K. and R.B.M; resources, A.S.; data curation, E.L.; writing—original draft preparation, S.R.G.; writing—review and editing, S.R.G., G.K, R.B.M, E.L; visualization, S.R.G., G.K; supervision, A.S. and G.K.; project administration, A.S.; funding acquisition, A.S.

**Funding:** This study was funded by the USDA NIFA national Integrated Water Quality Program under grant no. 2013-51130-21484 and by the National Science Foundation under grant No. OIA-1301789.

**Acknowledgments:** One of the authors of this article, Art Stoecker, passed away before submitting this work. The rest of the authors would like to express their gratitude and admiration to him and also expect that this article serves as a tribute to his memory.

**Conflicts of Interest:** The authors declare no conflict of interest.

## Appendix A

**Table A1.** Conventional or reduced tillage for dryland crops and pasture in the study area.

| Crop | Date (Month/Day) | Operation |
|---|---|---|
| **Cotton** | 1/1 | Tillage operation (Disk Plow Ge23ft) |
| | 3/15 | Tillage operation (Disk Plow Ge23ft) |
| | 5/15 | Tillage operation (Springtooth Harrow Ge15ft) |
| | 6/1 | Tillage operation (Finishing Harrow Lt15ft) Pesticide Operation (Pendimehalin, 0.25 kg) |
| | 6/10 | Fertilizer application (Elemental Nitrogen, 50 kg) |
| | 6/11 | Plant |
| | 7/1 | Tillage operation (Row Cultivator Ge15ft) |
| | 11/15 | Harvest and kill |
| **Pasture** | 1/1 | Plant |
| | 3/1 | Auto fertilization |
| | 5/1 | Grazing operation (Beef-Fresh Manure, GRZ_DAYS *: 180, BIO_EAT *: 3, BIO_TRMP *: 0.47, MANURE_KG *: 1.5) |
| **Winter wheat** | 3/15 | Fertilizer application (Elemental Nitrogen, 80 kg) |
| | 6/1 | Harvest and kill |
| | 7/1 | Tillage operation (Chisel Plow Gt15ft) |
| | 8/1 | Tillage operation (Offset Dis/heavduty Ge19ft) |
| | 9/20 | Fertilizer application (Elemental Nitrogen, 80 kg) (Elemental Phosphorus, 35 kg) |
| | 9/22 | Tillage operation (Disk Plow Ge23ft) |
| | 9/24 | Tillage operation (Springtooth Harrow Lt15ft) |
| | 9/25 | Plant |
| | 12/1 | Grazing operation (GRZ_DAYS *: 90, BIO_EAT *: 3, BIO_TRMP *: 0.47, MANURE_KG *: 1.5) |

**Table A1.** *Cont.*

| Crop | Date (Month/Day) | Operation |
|---|---|---|
| **Grain sorghum** | 5/1 | Plant |
| | 5/27 | Fertilizer application (Elemental Nitrogen, 150 kg) |
| | 5/28 | Tillage operation (Springtooth Harrow Ge15ft, Disk Plow Ge23ft, Mecoprop Amine, 125), Pesticide Operation (Mecoprop Amine, 125 kg) |
| | 10/18 | Tillage operation (Disk Plow Ge23ft) |
| | 10/20 | Tillage operation (Springtooth Harrow Ge15ft) |
| | 10/30 | Harvest and kill |
| **Alfalfa** | 4/1 | Harvest only |
| | 5/15 | Harvest only |
| | 7/1 | Harvest only |
| | 8/29 | Fertilizer application (Elemental Nitrogen, 50 kg), (Elemental Phosphorous, 20 kg) |
| | 9/7 | Plant |
| | 10/15 | Harvest only |
| **Hay** | 4/1 | Harvest only |
| | 7.1 | Harvest only |
| | 8/29 | Auto fertilization |
| | 9/7 | Plant |
| | 10/15 | Harvest only |
| **Rye** | 6/10 | Harvest only |
| | 8/10 | Fertilizer application (Elemental Nitrogen, 80 kg), (Elemental Phosphorous, 35 kg) |
| | 9/20 | Plant |
| | 9/15 | Grazing operation (GRZ_DAYS *: 150, BIO_EAT *: 3, BIO_TRMP *: 0.47, MANURE_KG *: 1.5) |

Note: * AUTO_NSTRS: Nitrogen stress factor of cover/plant triggers fertilization. This factor ranges from 0.0 to 1.0 where 0.0 indicates there is no growth of the plant due to nitrogen stress and 1.0 indicates there4 is no reduction of plant growth due to nitrogen stress; * GRZ_DAYS: Number of consecutive days grazing takes place in the HRU; * BIO_EAT: dry weight of biomass consumed daily ((kg/ha)/day); * BIO_TRMP: dry weight of biomass trampled daily ((kg/ha)/day); * MANURE_KG: dry weight of manure deposited daily ((kg/ha)/day).

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
