# Peer review of "Projected Climate Could Increase Water Yield and Cotton Yield but Decrease Winter Wheat and Sorghum Yield in an Agricultural Watershed in Oklahoma"

_water, doi:10.3390/w11010105_

Round 1
Reviewer 1 Report
Overall, your article reads Ok. However, it lacks structure and more major details. More critical analysis and more explanations are needed.
Comparisons with other SWAT studies are required in introduction and in conclusions sections. In addition, in the introduction, you should talk more about the use of SWAT models and their use.
Line 69-81, Section 2. Materials and Methods: All of this section should be deleted and shoud be written from scratch, again, properly to clearly describe the methodology you have used.
Daggupati et al (2018) state that °Large scale watersheds contributing to Lake Erie from the USA side are being simulated using hydrological and water quality (H/WQ) models such as the Soil and Water Assessment Tool (SWAT) and the results from the model are being used by policy and decision makers to implement better management decisions to solve emerging phosphorus issues.° (https://www.mdpi.com/2073-4441/10/2/222/htm)
Akkari and Bryant (2017) mention some modeling studies that are useful to compare your study with: https://www.mdpi.com/2077-0472/7/7/54/htm
Authors stated the following °In this study, we investigated climate change impacts on hydrology and crop yield in an agricultural watershed of Oklahoma using a SWAT based hydrological model°... since hydrology is a complex process, authors should specify what parts of hydrology they studied, in relation to climate change impacts. Future climate models should account the effects of phosphorous legacy.
Authors should talk about pros and cons of using such models, and suggest future recommendations, such as:
Cotton requires lots of water (and additional nutrients), so how will future water demands will be able to meet those demands in the face of increased population demands for potable water. Stating that
Overall, more background research is needed. For instance, Easterling et al., 1992; Rosenberg, 1992; Carter et al., 1994; Smit et al., 1996 state that °Increasing attention has been given to the prospects of farm-level adaptation to changed – and annual variable – climatic conditions, instead of focusing on plant growth and crop yields under long-term climate average climate scenarios,° Where does your research fit in this statement ?
Author Response
Reviewer 1.1 Overall, your article reads Ok. However, it lacks structure and more major details. More critical analysis and more explanations are needed.
Author Response: We would like to thank you for taking time to review our manuscript. We have addressed your concerns and included your suggestions in the revised version. In the revised version, you will see added information in the Introduction and Discussion sections.
Reviewer 1.2 Comparisons with other SWAT studies are required in introduction and in conclusions sections. In addition, in the introduction, you should talk more about the use of SWAT models and their use.
Author Response: We partially agree with your suggestion. We have used about a dozen SWAT model related studies in the methods, results and discussion sections. We like to keep the conclusion section as it is because we want to make sure we highlight the results of our study to be effectively presented and highlighted in the conclusion rather than comparing and contrasting our findings with other studies. In the introduction section, we highlighted crop yield and water quality-quantity impacts of climate change with several examples (such as, Ray et al. 2015; Kang et al. 2009; Kharel and Kirilenko, 2018). We have utilized the results of SWAT and other relevant studies (for example, Qiao et al., 2017; Garbrecht et al., 2014; Neupane et al., 2015; Gautam et al., 2018; Rosenzweig et al, 2007.; Delphine et al., 2014; Adhikari et al. 2016; Travis and Huisenga et al. 2013; Schattman et al, 2016) to discuss our findings in the Discussion section.
Reviewer 1.3 Line 69-81, Section 2. Materials and Methods: All of this section should be deleted and shoud be written from scratch, again, properly to clearly describe the methodology you have used.
Author Response: We believe that you read the pdf version of the manuscript in which the Section 2 has a default message to authors, which is not there in the word version of the submitted manuscript. Apart from this, if the Reviewer 1 has any additional concern in any specific component of our methodology, we would be more than happy to have an in-depth look into it and revise it as necessary.
We modified the manuscript structure by moving the Section 2.1 to the Section 2. Then, to reflect this change, we have renumbered the sections and sub-sections appropriately.
Reviewer 1.4 Daggupati et al (2018) state that °Large scale watersheds contributing to Lake Erie from the USA side are being simulated using hydrological and water quality (H/WQ) models such as the Soil and Water Assessment Tool (SWAT) and the results from the model are being used by policy and decision makers to implement better management decisions to solve emerging phosphorus issues.° (https://www.mdpi.com/2073-4441/10/2/222/htm)
Author Response: Daggupati et al. (2018) describe parameterizing a large-scale SWAT model for the entire watershed contributing to Lake Erie in USA and Canada. Several sources for important input data for the model (e.g. soil, landuse, and climate), and to achieve the desired model accuracy, several scenarios were developed that featured different combinations of these inputs. The paper offers interesting perspectives on model calibration and interpretation which were incorporated into the manuscript (See….). Similar to Daggupati et al. (2017), next steps for this study would be to use the Fort Cobb model for stakeholder education and management decision-making.
Reviewer 1.5 Akkari and Bryant (2017) mention some modeling studies that are useful to compare your study with: https://www.mdpi.com/2077-0472/7/7/54/htm
Author Response: Akkari and Bryant (2017) studied the factors affecting the adoption of nutrient-management BMPs by farmers. The studies that were cited by Akkari and Bryant (2017) and mentioned by Reviewer 1 fall into the general categories ‘nutrient BMPs’, ‘adoption of BMPs’, ‘farms as resilient human-environment systems’, and ‘knowledge dissemination of agricultural innovation’. Our paper is a general study focused on the hydrologic and crop yield effects likely to result with the precipitation and temperature projections of several GCM/RCP scenarios. Our study assumes steady-state crop distribution and management, so that the only changes will derive from climatic changes. Modeling optimal crop choice and management is a valuable ‘next step’, and those references would be important at that time. We have included this implication in the revised discussion section of the manuscript.
Reviewer 1.6 Authors stated the following °In this study, we investigated climate change impacts on hydrology and crop yield in an agricultural watershed of Oklahoma using a SWAT based hydrological model°... since hydrology is a complex process, authors should specify what parts of hydrology they studied, in relation to climate change impacts. Future climate models should account the effects of phosphorous legacy.
Author Response: Thank you for your useful comment. Yes, we acknowledge that the term ‘hydrology’ would be too vague for readers to understand what exactly this paper is focusing on. Therefore, we have replaced “hydrology” with “surface runoff” and “water yield” in the Section 5 and throughout the paper as needed. Regarding the second comment related to the “effects of phosphorous legacy” and climate models, our study only used future climate data projected by three GCMs and three RCPs and downscaled to our study region as described in the Section 2.3 (previously Section 2.4). Also, in this study, we didn’t investigate the impacts of climate change on nutrients loading. Therefore, addressing this comment is beyond the scope of our study.
Reviewer 1.7 Authors should talk about pros and cons of using such models, and suggest future recommendations, such as:
Author Response: We agree with your comment and added a new paragraph in the discussion section highlighting the limitations and future direction of this study as follow:
“Our study has important limitations that need to be addressed in the future research. The first limitation is related to modeling of sediment and nutrient loadings which we excluded in this study but are important in the selection and implementation of best management practices in agricultural watersheds. Next limitation is related to hydrological and climate model related uncertainty in the estimated water and crop yield. In this study, we accounted for climate model related uncertainty by including the ensemble of three GCMs and three RCPs as suggested by Brown et al. 2012”
Reviewer 1.8 Cotton requires lots of water (and additional nutrients), so how will future water demands will be able to meet those demands in the face of increased population demands for potable water. Stating that
Author Response: In this paper, we didn’t include the impacts of climate change on water quality, which we have discussed as one of our study limitations and should be included in the future research. However, we have clearly stated in the discussion that we used the historical irrigation trend for cotton to estimate cotton yield and corresponding water yield under the projected future climate. Also, we have noted in the manuscript (Section 2.1, previously section 2.2) that cotton is only 9% of the study area as compared to winter wheat which is 34%. The source of water for cotton irrigation is groundwater, and therefore we stated that there would be an opportunity for increased groundwater recharge in the region under the projected climate. In summary, in this study, our goal was not to estimate the potable water and water quality rather to estimate the impacts of climate change on surface runoff, water yield and crop yield.
Reviewer 1.9 Overall, more background research is needed. For instance, Easterling et al., 1992; Rosenberg, 1992; Carter et al., 1994; Smit et al., 1996 state that °Increasing attention has been given to the prospects of farm-level adaptation to changed – and annual variable – climatic conditions, instead of focusing on plant growth and crop yields under long-term climate average climate scenarios, Where does your research fit in this statement?
Author Response: Thank you for providing us with these important publications related to climate change impact studies. While these publications have laid out important points and direction for climate related research, the scope of our study is different. We didn’t perform experimental investigations rather used physically based hydrological model to study the impacts of changes in precipitation and temperature on yields of water and crops. In our SWAT model, the parameters for plant growth are well accounted for using planting and harvesting dates, fertilizer use, irrigation schedule, and management practices for specific crops that we have included in Appendix 1.

Reviewer 2 Report
By using the SWAT model and taking observed climate and streamflow into consideration, in this study, CMIP-5 projected climate was evaluated in terms of its impacts on hydrology and crop yields.
The research was well-designed and the introduction provided sufficient background. The paper was well organized and written. The authors reviewed relevant literature to support their research. The methodology, the analysis, the results and the conclusion were all clearly described.
The modeling and the predicting of the future climate is of high uncertainty. This study used certain future climate projections to predict. The accuracy of the results and the corresponding conclusions will affects the implications and applications, so as to the contributions of this paper.
Please provide the full names as the abbreviations first appear in the context. And also please re-check the written in the all article to detect some bugs.

Author Response
Synopsis: By using the SWAT model and taking observed climate and streamflow into consideration, in this study, CMIP-5 projected climate was evaluated in terms of its impacts on hydrology and crop yields.
Reviewer 2.1 The research was well-designed and the introduction provided sufficient background. The paper was well organized and written. The authors reviewed relevant literature to support their research. The methodology, the analysis, the results and the conclusion were all clearly described.
Reviewer 2.2 The modeling and the predicting of the future climate is of high uncertainty. This study used certain future climate projections to predict. The accuracy of the results and the corresponding conclusions will affect the implications and applications, so as to the contributions of this paper.
Author Response: We would like to thank you for your time reviewing our paper. We are glad that you liked our study.
Reviewer 2.3 Please provide the full names as the abbreviations first appear in the context. And also please re-check the written in the all article to detect some bugs.
Author Response: Thank you for spotting this particular issue of ‘abbreviations’. We have revised the manuscript and made appropriate changes to address your concern.

Reviewer 3 Report
I have included my comments in the pdf directly. The quality of the manuscript can be strengthened by incorporating uncertainty analyses.

Author Response
Reviewer 3.1 Line 19: explicitly mention that this is the projected climate data based on the downscaled Coupled Model Intercomparison Project 5.
Author Response: Thank you for your suggestion, which we agree with and we have modified this sentence as follow:
“This study estimated the potential impacts of the projected precipitation and temperature based on the downscaled Coupled Model Intercomparison Project 5 (CMIP-5) on hydrology and crop yield of a rural watershed in Oklahoma, USA”
Reviewer 3.2 Line 20: CMIP-5
Author Response: We inserted CMIP-5 in parenthesis as you suggested.
Reviewer 3.3 Ln 84 - 85 - consider paraphrasing this sentence.
Author Response: We modified the sentence as
“These two sub-watersheds were integrated into a single study area.”
Reviewer 3.4 Line 91: what is the difference between major and popular?
Author Response: We removed the term ‘popular’ in the revised manuscript.
Reviewer 3.5 Ln 107 - why not 3 m DEM data?
Author Response: It’s a good question. Initially, we started with two meter DEM that we have available locally but due to the computational overload and space issue resulted from 15,000 plus Hydrologic Response Units (combinations of high resolution SSURGO soil, land use and slope), we had to use the 10 m DEM. The delineation of the watershed and creation of the stream network had no changes between the use of either 10 meter or 2-meter DEM.
Reviewer 3.6 ln 152 - 154 - check the language here.
Author Response: We modified these sentences as:
“The study area was then divided into 43 sub-basins with an average area of 8 km2 (min. 0.2 km2 and max. 28 km2). Soil attributes, including texture and moisture capacity were derived from the Soil Survey Geographic Database -SSURGO [29]. The crop data layer for the year 2014 [22] was used to identify the locations of each crop and land cover types in the study watersheds.”
Reviewer 3.7 ln 185 - 187 - Check for language.
Author Response: We modified the sentences in the revised manuscript as below:
“The model performance for streamflow was evaluated using three statistical measures: the coefficient of determination (R2), the Nash-Sutcliffe efficiency (NSE), and percentage bias (PB). The values of R2 (0.64), NSE (0.61), and PB (5%) (Figure 3) in the model calibration period were deemed to be satisfactory by metrics suggested by other SWAT-based studies [34,35].”
Reviewer 3.8 Ln 250 - 256. Not sure if I fully understood. are you trying to say that you incorporated the irrigation management practices in the model and then separated the difference in yield between irrigated and non-irrigated areas? Please clarify.
Author Response: Yes, we separated the irrigated and dry cotton areas by using the center-pivot locations from aerial imageries, and then applied irrigation schedule only to the areas identified as irrigated based on center-pivot locations. Yes, we obtained yields for both dry and irrigated cotton which we have included in the discussion.
Reviewer 3.9 Ln 262- Do we need to add this "Understanding of the potential future changes...."
Author Response: We rephrased the sentence as:
“The potential future changes in precipitation and temperature and how they interact to affect the water balance in agricultural watersheds is crucial.”

Round 2
Reviewer 1 Report
Thank you for replying on my comments. Your replies were clear, as a result I accept your article to be published in present form.
Author Response
First of all, we would like to thank you for reviewing our manuscript and providing us with constructive comments. We are glad that you were satisfied with our response to your comments and you accepted our revised manuscript for publication in Water.
Reviewer 3 Report
I have included my comments in the document.

Author Response
Comment 3.1: mention explicitly that these statistical matrices were for streamflow.
Response: We agree with your comment and therefore we edited the sentence as below:
Three statistical matrices: coefficient of determination (R2), Nash-Sutcliffe efficiency (NSE), and percentage bias (PB) were used to gauge the model performance with satisfactory values of R2 =0.64, NS =0.61, and PB=+5% in the calibration period and R2 =0.79, NSE=0.62, and PB=-15% in the validation period for streamflow.
Comment 3.2: This is bit subjective. You need to give the statistical matrices to support your claim of acceptable model performance.
Response: Thank you for this constructive comment. We agree with you and therefore included percent bias for each crop as below:
The model parametrization for the yields of cotton (PB=-4.5%), grain sorghum (PB=-27.3%) and winter wheat (PB=--6.0%) resulted into an acceptable model performance.
Comment 3.3: change the figure's background.
Response: The figures we submitted to the journal don’t have any background color. We assume that the background colors in figures are part of the formatting done by the journal technical team. Therefore, we would like to pass this comment to the journal to make appropriate changes as deemed necessary by the journal.
Comment 3.4: Ln 152 - 154. Check the language.
Response: In the revised manuscript, we changed the sentence as below:
Manual calibration of the model for yields of cotton, grain sorghum and winter wheat followed successful hydrologic calibration using percent bias as a measure of performance which is popularly used in crop-yield modeling studies [41,42].
Comment 3.5: change the figure background to white. Also I suggest including crop's name before yield in the y axes. This way, readers don't have to completely rely on figures' legend to understand which figure is for which crop.
Response: Regarding the figure background color, please refer to our response above – comment 3.3. We agree with your suggestion and therefore added crop name in Y-axis of Figure 4a, 4b, and 4c.
Comment 3.6: There are several downscaled future climate projections data. Why did you select this particular one? What is your rationale behind selecting only few GCM's when there are several GCMs.
Response: Yes, we agree with you that there are several downscaled climate projection data sources. However, we selected the product made ready by the United States Geological Survey – South Central Climate Science Center (SCCSC) because these data were specifically prepared for the Southern Great Plains region in which our study watersheds are located. These climate data are downscaled and bias-corrected for this region therefore the accuracy of these data is high. We used all three GCMs that are made available by the SCCSC. The process of GCM selection and downscaling methods are described in http://dx.doi.org/10.15763/DBS.SCCSC.RR. The section 2.3 in the manuscript clearly answers your comment.
Comment 3.7: Again, change the figures's background. Please do this for the following figures also.
Response: Please see above in our response to your comment 3.3 for clarification.
